# Deep Learning-Based Denoising of CEST MR Data: A Feasibility Study on Applying Synthetic Phantoms in Medical Imaging

**DOI:** 10.3390/diagnostics13213326

**Published:** 2023-10-27

**Authors:** Karl Ludger Radke, Benedikt Kamp, Vibhu Adriaenssens, Julia Stabinska, Patrik Gallinnis, Hans-Jörg Wittsack, Gerald Antoch, Anja Müller-Lutz

**Affiliations:** 1Department of Diagnostic and Interventional Radiology, Medical Faculty, University Dusseldorf, 40225 Dusseldorf, Germanyantoch@med.uni-duesseldorf.de (G.A.); anja.lutz@med.uni-duesseldorf.de (A.M.-L.); 2F.M. Kirby Research Center for Functional Brain Imaging, Kennedy Krieger Institute, Baltimore, MD 21205, USA; 3Division of MR Research, The Russell H. Morgan Department of Radiology and Radiological Science, The Johns Hopkins University School of Medicine, Baltimore, MD 21205, USA

**Keywords:** CEST, deep learning, synthetic phantoms, noise detection, noise reduction, noise suppression

## Abstract

Chemical Exchange Saturation Transfer (CEST) magnetic resonance imaging (MRI) provides a novel method for analyzing biomolecule concentrations in tissues without exogenous contrast agents. Despite its potential, achieving a high signal-to-noise ratio (SNR) is imperative for detecting small CEST effects. Traditional metrics such as Magnetization Transfer Ratio Asymmetry (MTR_asym_) and Lorentzian analyses are vulnerable to image noise, hampering their precision in quantitative concentration estimations. Recent noise-reduction algorithms like principal component analysis (PCA), nonlocal mean filtering (NLM), and block matching combined with 3D filtering (BM3D) have shown promise, as there is a burgeoning interest in the utilization of neural networks (NNs), particularly autoencoders, for imaging denoising. This study uses the Bloch–McConnell equations, which allow for the synthetic generation of CEST images and explores NNs efficacy in denoising these images. Using synthetically generated phantoms, autoencoders were created, and their performance was compared with traditional denoising methods using various datasets. The results underscored the superior performance of NNs, notably the ResUNet architectures, in noise identification and abatement compared to analytical approaches across a wide noise gamut. This superiority was particularly pronounced at elevated noise intensities in the in vitro data. Notably, the neural architectures significantly improved the PSNR values, achieving up to 35.0, while some traditional methods struggled, especially in low-noise reduction scenarios. However, the application to the in vivo data presented challenges due to varying noise profiles. This study accentuates the potential of NNs as robust denoising tools, but their translation to clinical settings warrants further investigation.

## 1. Introduction

Chemical Exchange Saturation Transfer (CEST) imaging has been recognized as a pivotal tool in the realm of biosensitive magnetic resonance (MR) imaging (MRI) [1,2]. It allows for the comprehensive analysis of tissue biomolecule concentrations without the need for exogenous contrast agents [3]. Although CEST imaging can provide valuable information about solutes at low concentrations, a high signal-to-noise ratio (SNR) is essential for accurately detecting subtle CEST effects [4,5]. Achieving a high SNR ensures that the CEST effects, no matter how subtle, are distinctly discernible against the background noise. This becomes especially crucial when working with low concentrations or when the effects are inherently small, as a compromised SNR could lead to potential misinterpretations or even missed detections.

While conventional methodologies, including Magnetization Transfer Ratio Asymmetry (MTR_asym_) and Lorentzian analyses, are effective metrics for CEST imaging [6,7], they do exhibit susceptibility to image noise [5,8]. Mathematically, MTR_asym_ is defined as the difference between the signals at positive and negative frequency offsets with respect to the water resonance. This metric essentially compares the two sides of the Z spectrum. The symmetry of the Z spectrum is disturbed by the presence of noise, leading MTR_asym_ to provide skewed results. Lorentzian analysis involves fitting the CEST spectrum to Lorentzian line shapes, where noise in the spectrum can invalidate this assumption especially since the peaks are no longer symmetrical.

This susceptibility inherently lowers the precision, complicating the shift to quantitative concentration estimations by CEST in the in vivo analyses [5]. However, one must acknowledge the omnipresence of noise across all image-processing modalities. In response, a slew of noise-reduction algorithms have been introduced in recent years, notable among which are principal component analysis (PCA), which allows noise reduction based on ordering noise and signal components in the Z-spectrum signal curve [8,9], nonlocal mean filtering (NLM), which is based on a weighted signal adjustment of the nearest neighbors [7,10], and block matching combined with 3D filtering (BM3D), which associates 2D blocks of the image with similar areas in the image to reduce noise [10,11,12].

In recent years, neural networks (NNs) have been applied in many areas, such as segmentation [13,14], classification [15], and also noise reduction [16,17,18]. While there are already studies like the one by Hunger et al. that have shown the potential of Deep Learning (DL) for CEST imaging [19], to our knowledge, the potential for noise reduction has not yet been investigated. For techniques like Diffusion Tensor Imaging (DTI), the feasibility of noise reduction using DL has already been demonstrated [20]. These networks, consisting of an encoder and a decoder, are characterized by their structure and operation as effective means for noise reduction [21,22].

Nevertheless, the efficacy of these NNs is invariably tethered to the volume and quality of the training data on hand. In medical imaging, challenges such as patient privacy concerns, data variability, rarity of certain conditions, and the need for expert annotations make procuring pertinent data particularly difficult [13,23,24]. However, the Bloch–McConnell equations offer an optimistic perspective. With these equations, the signal response inherent in CEST images can be determined numerically, paving the way for the generation of synthetic MR images [5,6,25,26].

In this work, we analyze the use and efficiency of NNs in CEST imaging, focusing mainly on applying autoencoders. Based on synthetically generated phantoms, we aimed to develop autoencoders for the noise reduction in CEST images. We compared the performance of these neural architectures with established analytical image denoising methods such as PCA, BM3D, and NLM based on simulated anatomical data, in vitro phantom measurements, and an in vivo intervertebral disc (IVD) measurement. Our hypotheses are

(1)Neural Networks, especially autoencoders, are potentially superior in denoising CEST images compared to traditional denoising methods.(2)Given adequate training data, NNs can consistently detect and suppress noise more efficiently than analytical algorithms.(3)The models trained in this study can be applied effectively to real CEST data.

For this purpose, this study generates synthetic data, trains NNs based on this data, and validates the performance on the anatomical Zubal phantom, phantom measurements, and in vivo data.

## 2. Materials and Methods

### 2.1. Study Design

The present work was designed as a prospective feasibility study. It includes sequential in silico, in vitro, and in vivo CEST MRI examinations performed in the following order: (1) implementation and validation of an autoencoder for denoising in silico data, (2) validation of the developed approach using in vitro experiments, (3) evaluation of in vivo transferability of an in silico-developed autoencoder on IVDs.

Written informed consent was obtained from the female volunteer and the study was approved by the local ethics committee (Ethical Committee, Medical Faculty, University of Düsseldorf, Germany, study number 5087R).

### 2.2. Generation of In Silico Phantoms

First, we created a synthetic CEST dataset to serve as a reference standard (similar to ground truth) for improving the NN noise reduction methods. This was achieved by creating in silico phantoms and combining various geometric shapes of different sizes and arrangements. Over 2000 iterations of layering these shapes, along with morphological optimization, resulted in complex geometric phantoms with smoothed outlines, as depicted in Figure 1.

In addition, an ellipsoidal foreground with morphological structures was created to simulate the diversity of biological structures. We created specific distribution maps for each defined pool and the associated parameters (Table 1). These maps were integrated with Pool A, symbolized by water, to uniformly depict geometric shapes and water signals.

The foundation of our synthetic image generation lies in the Bloch–McConnell equations, a set of differential equations that describe the evolution of nuclear magnetization in a multi-pool exchange system under the influence of radiofrequency irradiation. In the context of CEST MRI, these equations allow for an accurate representation of the magnetization transfer between the solute and solvent pools. The capacity to simulate these exchange dynamics provides a nuanced understanding of CEST contrast mechanisms, making it possible to generate realistic synthetic images that replicate the intricacies of actual CEST MRI scenarios. Utilizing the Bloch–McConnell equations, we extrapolated the CEST exchange dynamics across discrete spin systems. By employing the created phantoms along with the set thresholds for CEST and relevant MR parameters (Table 1), we generated a collection of 10,000 random 2D CEST datasets, each with a resolution matrix of 128 × 128 pixels and including 50 offset frequencies. The Z spectra for each dataset were customized at a voxel level, encompassing a spectrum of 2–5 pool systems, including a singular water reservoir and up to a quartet of exchange pools. We used Gaussian saturation pulses throughout the computational process, reflecting our standard clinical MRI protocols [5,27]. However, parameters such as echo time (TE), repetition time (TR), and offset frequency range (Δω) were subjected to variations. The digital framework we developed, available at [GitHub: https://github.com/MPR-UKD/CEST-Generator, last access on: 7 May 2023], is based on the Bloch–McConnell Simulation Tool by Zaiss et al. (accessible at https://github.com/cest-sources/BM_sim_fit/, last access on: 7 May 2023) and is made available to the scientific community under a GNU license. The Bloch–McConnell simulation by Zaiss et al. has already been validated in numerous studies [25,26] and allowed a transfer from the in silico experiments to the in vivo studies [5,6]. With our extension, 2D images can be simulated, offering new potential for further studies.

### 2.3. Neural Network Architectural Design

The designed models are based on the UNet architecture and its modified form, the ResUNet (Figure 2). Both architectures are characterized by an encoder, a latent space, and a decoder [22,28].

The UNet and its modification, ResUNet, are deep convolutional neural network architectures that have shown significant prowess in medical image processing tasks, especially in image denoising. Their design, which conserves spatial context throughout the encoding and decoding phases, makes them inherently adept for denoising applications. The ResUNet, with its enhanced structure featuring additional residual blocks, facilitates an efficient flow of information through the network. This ensures a meticulous suppression of noise while preserving the essential features of the image [29].

In the UNet encoder, a sequence of “down” blocks extracts abstract representations of the input data x∈X, and at the same time, the dimensionality is reduced. Each “down” block processes the input via a sequence of 2D convolution layers (Figure 2: left path).

The latent space further transforms the data using two residual blocks, doubling the number of features: $z=R_{l}\left(F\left(x\right)\right)$, where $R_{l}$ represents the residual block in the latent space [30].

The decoder of the UNet uses “Up” modules to gradually increase the spatial resolution (Figure 2: right path). Each “Up” module U can be represented as $U\left(z\right)=G\left(z\right)⊕z$, where $G\left(z\right)$ symbolizes the transformation via the decoder and ⊕ signifies the concatenation of features [28].

The ResUNet extends the UNet using additional residual blocks in the encoder and decoder. These blocks allow input information to be passed directly to the output layers, minimizing information loss during training [22,30]. A residual block $R_{d}$ can be described as $R_{d}\left(x\right)=F\left(x\right)+x,$ where $F\left(x\right)$ represents the transformation via two successive 2D convolution layers. The 1x1 convolution, which serves as a shortcut, allows for channel matching between the input and output [13,31].

For both proposed models, namely the U-Net and its counterpart ResUNet, a dual-pronged training strategy was implemented. The first model $M_{1}$ was trained to estimate the noise and subtract it from the image x: $x^{\prime }=x-M_{1}\left(x\right)$. The second model $M_{2}$ was trained to remove the noise directly from the image data, $x^{\prime }=M_{2}\left(x\right)$. For ease of reference in the subsequent sections, the models are denominated as “ResUNet-NE-yes” when the ResUNet incorporates noise estimation and “ResUNet-NE-no” when it omits this feature. A parallel nomenclature will be adopted for the U-Net variants.

To guarantee each structure achieves the best middle ground between intricacy and capability, their depth was set at four levels. This provided a sufficient abstraction gradient which is crucial for the effective extraction of relevant data features including noise assessment and its following reduction.

### 2.4. Training

The computational experiments were executed on a specialized workstation, fortified with dual Intel^®^Xeon^®^Gold 6242R CPUs (Intel Corporation, Santa Clara, CA, USA) and an expansive memory allocation of 376 GB RAM. For the purpose of expediting the training and computational tasks, the infrastructure incorporated four RTX 3090 GPUs (NVIDIA, Santa Clara, CA, USA), facilitating parallelized training across the entirety of the graphical processing units. The computational framework was constructed upon Python (version 3.10), with the architectural designs of the models being instantiated via PyTorch v1.13.0 [32] and further enhanced using PyTorch Lightning [33]. Model tuning was performed using the ADAM optimization algorithm initialized with a learning rate parameter of 0.01 and supplemented by a weight loss coefficient of 10^−6^ [34]. To ensure adaptive learning rate modulation throughout the training trajectory, a scheduler mechanism (“ReduceLROnPlateau”) was integrated, characterized by a patience parameter of three and a decrement factor of 0.1. The training was over 30 epochs. The selected loss metric was the Mean Squared Error (MSE) and the data batches were partitioned into batches of 40. Mimicking real operational conditions, each image during training was exposed to a variable noise parameter sigma ranging between 0 and 0.2. Initially, a Fourier transformation was applied to convert the image data to the frequency domain. Afterwards, Gaussian noise was added to both the real and imaginary components. A subsequent inverse Fourier transformation allowed the reconstruction of the image, incorporating the noise elements [35]. Once the noise was added, the altered image was fed into the model. Additionally, a random frequency deviation was applied, extracting a sequence of 41 dynamics from the original set of 50 dynamics.

### 2.5. Analytical Denoising Techniques

In this study, we compared the neural model’s denoising performance against analytical noise suppression methodologies. These techniques encompassed non-local means (NLM), principal component analysis (PCA), and block matching combined with 3D filtering (BM3D).

PCA: The PCA technique aims to denoise the multidimensional dataset D by retaining only the principal components with the highest variance [8,9]. Mathematically, the dataset D is transformed into a set of orthogonal vectors, with each vector representing a principal component. The decision on the number of principal components to retain is guided by criteria such as “median”, “Nelson”, and “Malinowski” [8], where k represents the optimal number of principal components:(1)$$D^{\prime }=PCA_{k}\left(D\right)$$

Naming Convention: For instance, a PCA using the Nelson criteria would be named PCA-Nelson.

BM3D: The BM3D technique, based on the work of Dabov et al. [36], is an advanced image-denoising method characterized by its high efficiency and quality [37]. The algorithm has two main phases: Block matching and 3D filtering [36], of which these two main phases are run through twice to obtain a baseline estimate and then the denoised image. During the block-matching phase, the algorithm searches for blocks similar to each reference block within the image. These analogous blocks are then arranged in a 3D array, with the number of similar blocks defining the third dimension. Mathematically, this process can be depicted using the mapping function M, which, given the reference block $B_{r}$, yields a stack of similar blocks S [36,37]:(2)$$S=M\left(B_{r}\right)$$

In the 3D filtering phase, this 3D block stack is transformed, typically with a 3D transform function T; in this work, it is the 3D discrete cosine transform. After the transformation, the coefficients in the transform space are filtered to reduce noise. Then, an inverse 3D transform is performed to return the filtered block stack to the image space. This can be mathematically described as
(3)$$S^{\prime }=T^{-1}\left(F\left(T\left(S\right)\right)\right)$$
where F is the filter function in the transformation space. The optimization in BM3D includes the variation in the 2D block size B and the search window size N. By varying these parameters, a balance between noise reduction and the preservation of image information is sought. These parameters have a decisive influence on the computation time of the algorithm.

Naming Convention: For instance, a BM3D configuration with a window size of 11 and block size of 4 would be denoted as BM3D WS_11_BS_4.

NLM: NLM is an adaptive filtering method that exploits image redundancy to mitigate noise [7,38,39]. For any pixel p and its associated search window W, the weight of neighboring pixels is determined using the likeness of their local surroundings. This relationship can be expressed:(4)$$p^{\prime }=NLM_{W}\left(p\right)$$

The dimension of the search window W directly impacts noise suppression. A broader W enhances noise reduction, albeit with the trade-off of potential image blurring. The NLM algorithm employs two key window sizes: A larger “search” window and a smaller “local” window.

Naming Convention: For instance, a configuration using a big window of 21 and a small window of 5 is denoted as NLM BW_21_SW_5.

### 2.6. In Silico Validation

Digital phantom: To investigate the properties of digital phantoms in the presence of noise, two specific Zubal phantoms [40], specifically of the abdomen and head, were used (Figure 3). These phantoms are publicly available from the Image Processing and Analysis Group at Yale University (https://noodle.med.yale.edu/zubal/, last access on: 20 August 2023). In each defined region of these phantoms, different generated Z spectra were superimposed based on Bloch–McConnell simulations and using the same parameters as the neural training phantoms. A total of 200 such customized phantoms were created, with an equal distribution of 100 each for the head and abdomen phantoms. Considering the three-dimensional nature of these phantoms, a single, random layer was selected for analytical examination during each iteration.

Analysis: The collection of 200 CEST datasets was systematically contaminated with different noise levels, indicated by sigma values from 0.01 to 0.25 (0.01, 0.02, 0.03, 0.04, 0.05, 0.075, 0.1, 0.15, 0.2, and 0.25). After the introduction of this noise, the datasets underwent detailed analysis and processing using various denoising methods. The peak signal-to-noise ratio (PSNR) of the Z spectra was used as a quantitative measure to evaluate the effectiveness of each denoising method. PSNR can be mathematically expressed as
$$PSNR=10\cdot \log_{10}\left(\frac{{MAX}_{I}^{2}}{MSE}\right)$$
where *MAX_I_* is the maximum possible pixel value on the image (1 for the Z spectra) and *MSE* is the mean squared error. This metric offers an impartial benchmark to ascertain the caliber of the denoised data vis-à-vis the unadulterated signal.

### 2.7. MRI Validation

All MRI evaluations were performed on a 3T MRI (MAGNETOM Prisma, Siemens Healthineers, Erlangen, Germany). Depending on the specific examination, either a dedicated 15-channel knee coil (Tx/Rx Knee 15 Flare Coil, Siemens Healthineers, Erlangen, Germany) for the phantom CEST study or a 32-channel body coil (Siemens Healthineers, Erlangen, Germany) for the in vivo CEST study was employed. MRI protocols comprised a set of localization sequences tailored for planning and a region-specific CEST measurement, detailed in Table 2. Mirroring the design of the simulated phantoms, a series of 42 presaturated images were captured at distinct saturation frequencies, all symmetrically centered around the water resonance, with a reference image at 300 ppm.

#### 2.7.1. In Vitro Study

Phantom Construction: An MR-compatible phantom with slots for eight test tubes was used, as outlined in a previous study [6]. Creatine (Carl ROTH, Karlsruhe, Germany) was introduced into these tubes at progressive concentrations: 50 mM, 100 mM, 150 mM, and 200 mM. This was achieved by dissolving the creatine in a PBS (phosphate-buffered saline) solution stabilized to a pH of 7.3. Tubes located across from each other in the phantom held the same concentration (Figure 4).

In vitro phantom measurement: After preparation, the phantom was placed at the center of both the knee coil and the MR scanner. Subsequently, two separate recordings of the phantom were initiated using the same CEST sequence in terms of saturation time, echo time, and base resolution. The first measurement, considered as the reference or “ground truth,” had a slice thickness of 10 mm, whereas the second had a slice thickness of 1.5 mm, to introduce a higher noise profile.

Evaluation: Post-measurement, both datasets underwent evaluations, encompassing their original state and denoised versions. In this evaluation, a Python-based algorithm was utilized to calculate the MTR_asym_ values, focusing specifically on the creatine-relevant frequency range of 1.5–2.5 ppm. The Water saturation shift referencing (WASSR) the correction was applied to the denoised CEST signal for B_0_ accuracy [41]. Following the correction, MTR_asym_ maps were generated and the mean as well as the standard deviation for each tube were calculated.

#### 2.7.2. In Vivo Studies

In vivo measurement: To investigate the transferability of our trained deep learning models to the in vivo data, a 26-year-old female volunteer was positioned feet-first in the supine position in the MR scanner. A body coil was placed at the level of the lumbar spine, while the spine coil integrated into the MR scanner table was located below the back. The volunteer was positioned so that the lumbar spine was centrally located in the isocenter of the scanner. For the MR measurement, a B_1_ of 0.9 µT was used, analogous to previous studies [5,27].

Evaluation: The measurement was evaluated both without and with denoising. MTR_asym_ values were determined in the OH-specific range of glycosaminoglycans (0.9 ppm—1.9 ppm) [42]. Analogous to the phantom measurement, the Z spectra were self-corrected with WASSR and MTR_asym_ maps were generated.

### 2.8. Evaluation Metrics

The investigation of the trained autoencoders regarding their ability to denoise CEST images by compressing and reconstructing the image data was the focus of our study. Analogous to the analytical methods, a noisy Z spectrum is passed and a denoised Z spectrum is returned. To provide a reliable quantitative measure for evaluating the effectiveness of the various denoising strategies, the peak signal-to-noise ratio (PSNR) was used as the primary assessment criterion.

For the digital (in silico) phantom evaluations, PSNR acted as a critical tool to measure the accuracy of the reconstructed signal profiles on a pixel-by-pixel basis throughout each Z spectrum. This allowed for an unbiased assessment of the denoising methods’ ability to reproduce the noise-free signal characteristics.

For the MRI data, a visual inspection of the generated MTR_asym_ maps was prioritized. The focus of evaluation was on identifying a consistent signal pattern within individual tubes and detecting the clear boundary between different concentration levels and consistent values between IVDs were expected.

## 3. Results

### 3.1. Neural Network Training Assessment

Throughout the iterative training over 30 epochs, the NN architectures—UNet-NE-No, ResUNet-Ne-Yes, and UNet-NE-No—showed significant improvement in reconstructing the CEST signals. The benchmark PSNR for the test dataset was 20 without denoising, but the neural architectures improved this metric to 34.3, 35.0, and 32.9, respectively, indicating a considerable increase in signal accuracy. Contrarily, the UNet-NE-Yes architecture demonstrated only incremental enhancement in the signal-to-noise ratio, achieving a PSNR of 21.2. This suggests that this UNet variant largely replicated the original image without making significant contributions to the denoising process.

Furthermore, the final trained models, as well as the implemented denoising methods, can be used via a simple command line tool and are freely available on GitHub: https://github.com/MPR-UKD/CEST-Denoise, last access on: 7 May 2023.

### 3.2. Digital Phantom (In Silico) Analysis

The simulations showed a clear relationship between the denoising performance, the magnitude of the noise perturbation, and the denoising strategy used (Figure 5). When applying the PCA algorithm, both the “median” and “Malinowski” criteria led to a steady increase in PSNR values via the varying noise levels. This trend suggests that as the noise levels rose, the PCA methods attempted to retain more signal information. However, there was significant variance between the different layers of the phantom, which was reflected in substantial standard deviations. In contrast, the Nelson criterion provided PSNR results comparable to the noise, indicating excessive component retention, implying a potential overfitting in noise representation. Although the BM3D and NLM techniques also produced successful results as the noise level increased, they proved inapplicable, potentially due to their inherent design for more generic noise patterns. Notably, the NN models, which were effective during the training phase, continued to show commendable denoising capabilities, achieving PSNR ranges of 30–40. This highlights the adaptability and robustness of neural networks in handling diverse noise structures. Their performance distinctly outperformed the PCA-based methods, which was particularly evident above a noise intensity of 0.05, underscoring the superiority of trained neural models in challenging the noise conditions.

### 3.3. In Vitro Phantom Measurement

When studying the phantom with a slice thickness of 10 mm, a noise level (sigma) of 0.02 was measured and most of the denoising methods were effective in minimizing noise in the CEST MTR_asym_ values (Figure 6, left panel). However, deviations from this pattern were noted in the two UNet-based neural algorithms and the BM3D techniques with a window size of 37. The phantoms integrated with creatine concentrations manifested a progressive augmentation in the MTR_asym_ metrics, commencing at approximately 0.5% at 50 mM and culminating at a substantial 3.6% at 200 mM. In evaluations with a reduced slice thickness of 1.5 mm, an increase in the noise sigma = 0.24 was observed. In alignment with the trajectories delineated in prior in the in silico investigations, the ResUnet algorithms approximated the MTR_asym_ metrics ascertained for the 10 mm phantom under pronounced noise conditions, effectuating noise attenuation (Figure 6, right panel). While the PCA methods aided in noise reduction, discrepancies occurred at concentrations between 150 mM and 200 mM. Supporting the findings of the in silico analyses, the NLM method provided minimal noise suppression. In contrast, the BM3D approach struggled to effectively neutralize the noise within the MTR_asym_ dataset, leading to physiologically incompatible results.

### 3.4. In Lumbar IVD Evaluation

In vivo, the studies of the lumbar IVD quantified a noise parameter (sigma) of 0.15. PCA-based methods, especially those utilizing “median” and “Malinowski” criteria, achieved an improved MTR_asym_ effect within the IVD. In addition, the observed variability between IVD segments was subtly attenuated, enhancing the differentiation between the nucleus pulposus (NP) and annulus fibrosus (AF). In contrast, the NN methods (ResUnet and UNet) caused a significant reduction in MTR_asym_ effects beyond the anticipated physiological ranges, with the MTR_asym_ metrics displaying values below zero (Figure 7). Non-local means (NLM) techniques tended to homogenize the data, resulting in slightly attenuated CEST responses, but they were still consistent with the datasets without denoising. In line with prior in vitro evaluations, the BM3D method produced MTR_asym_ results that strayed from the physiological standards.

## 4. Discussion

In this study, we assessed the efficacy of various denoising methods, emphasizing the role of NNs in enhancing CEST images. Our findings underscore that the in silico-generated phantom data acts as a valuable resource for developing and fine-tuning neural methods that efficiently remove noise from CEST MRI images. Notably, the two distinct ResUNet methods proved particularly effective for the data contaminated by substantial noise from thermal processes, delivering results comparable to, if not superior to, analytical methods like PCA. The performance assessment of ResUNet on our synthetic phantoms revealed its significant capabilities in noise suppression, particularly for thermal process-related noise. Crucially, our results indicate that while traditional denoising methods offer consistent performance, neural network-based approaches, particularly ResUNet, demonstrate enhanced adaptability and precision in diverse and challenging noise scenarios. Among the evaluated neural architectures, ResUNet-NE-yes, which employs a two-step approach of noise identification followed by image combination, showcased a marginally superior performance over the direct denoising method. These outcomes reiterate the potential of neural networks, especially architectures like ResUNet, in enhancing CEST MRI images and further emphasize the instrumental role of in silico phantoms in refining and validating such neural methods. It is important to note that a method, which first identifies the noise and then combines it with the original image (referred to here as ResUNet-NE-yes), slightly excelled over the approach that exclusively reconstructs the denoised image. However, a significant difference was seen when comparing the in silico and in vitro datasets with the in vivo data. This divergence is primarily due to different noise distributions, which were not considered during the NN training phase but appeared in vivo. Visible movements in the image space due to respiration and peristalsis were not visible in the ROIs, but the NNs analyzed the whole image without a prior selection of ROIs.

Furthermore, the proficiency of noise reduction algorithms, especially when utilizing neural network-based strategies, plays a pivotal role in the quantitative estimation of biomolecule concentrations in tissues via CEST MRI. As concentration determinations typically depend on discerning differences in the MTR_asym_ values obtained at varied B_1_ field strengths, it becomes essential that the data remain consistent and free from noise-induced distortions across multiple measurements. With the enhanced image clarity offered by neural network denoising, we can ensure a more dependable extraction of these values, leading to accurate biomolecular concentration measurements. This advancement, in turn, magnifies the diagnostic and prognostic acumen of CEST MRI, solidifying its stature as a trusted tool in clinical studies. From an analytical denoising standpoint, the PCA method demonstrated consistent superiority, overshadowing the NLM and the BM3D techniques. This observation aligns with the foundational work presented by Breitling et al. [8], where the essence of segregating the signal into pertinent segments as opposed to noise-laden components was emphasized for CEST MR imaging. After this segmentation, noise elimination is simplified. A challenge, however, persists in determining the optimal number of components to ensure effective noise reduction. Mirroring observations from the aforementioned study by Breitling et al., our research confirmed that while the Nelson criterion showed less than ideal denoising effectiveness, both the Median and Malinowski criteria showed strong performance [8]. A notable challenge encountered during the study was the 1.5 mm in vitro phantom. Here, while the data was successfully denoised, the reduction in components curtailed the discernment between the concentrations of 150 mM and 200 mM creatine, a distinction that was clear in the baseline 10 mm measurement. Contrarily, the neural methodologies under investigation achieved robust noise reduction an almost impeccable MTR_asym_ distribution. However, for the in vivo evaluation of the IVD, the PCA methods were again convincing and allowed differences as described in the literature between NP and AF, where these differences were visible in the image. In contrast, in previous studies, they were only marginal and not always clearly visible [27,43].

The application of NLM as a denoising technique in MRI has been validated via numerous studies and its utility in CEST studies is also well documented [7,10,38,39]. In a parallel development, BM3D has surfaced as a modern extension, mainly in the field of image processing. However, our research findings indicated that both NLM and BM3D were not ideally suited for our specific CEST experiments. This observation is in contrast to a recent study by Romdhane et al., where the effective application of both BM3D and NLM to the CEST data was displayed [10]. However, in the study, Iopamidol was used to enhance the CEST effect and only a field of view of 3 cm was evaluated, where the pixels in the image therefore had higher spatial correlations than in our study and in clinical use. It is imperative to highlight that our MR measurements presented inherent challenges for these image-centric algorithms. For instance, in the phantom measurement, the subtle luminance variations between the creatine tubes and the adjacent water might render them indistinguishable. Similarly, in the lumbar spine measurement, the diminutive size of the discs juxtaposed with the surrounding tissue devoid of CEST effects posed specific challenges for BM3D, which operates on a global scale. This was evident in our findings. Concurrently, NLM’s performance was suboptimal, resulting in only slight noise reduction in our tests. Like BM3D, NLM considers the relationship with the nearest neighbors. At 3 Tesla, CEST effects typically are minimal [40,41].

The focus of this study was to investigate the capabilities of NNs in the complicated area of noise reduction in CEST datasets using only simulated data for the training process. The exchange of CEST processes can be described and simulated mathematically via the Bloch–McConnell equations [5,6,25,26]. This foundational theory was augmented using a systematic combination of synthetic phantoms, exchange rates, relaxation times, and concentrations, leading to an extensive collection of 10,000 phantoms, each calculated meticulously on a detailed, pixel-wise level. The inherent noise in MR images is mainly due to thermal fluctuations [44]. Such thermal noise, principally arising from the random motion of electrons within the coils, induces intrinsic signal fluctuations [45]. We modelled this by superimposing noise in the frequency space on both the real and imaginary signal. Afterwards, we successfully trained different model architectures over 30 epochs.

In our study, we were able to demonstrate the potential of NNs trained only on simulated data and the transfer to the in vitro data, but we also observed challenges for the in vivo data. The idealized noise types utilized during our training sessions did not appear to align with the complex noise environment encountered in the in vivo experiments. For example, variables such as respiratory- or intestinal movement-related signal fluctuations in the abdomen degrade the quality of our neural models, thus leading to misinterpretations. For our simulation experiments, at low noise intensity, the superiority of NNs over the PCA method remains inconclusive. However, as the noise intensity increased, the neural algorithms began to show pronounced effectiveness. For transferability to the phantom measurements, it was noticeable that only the ResUNet methods could accurately distinguish between concentrations of 150 mM and 200 mM of the phantom acquired with a slice thickness of 1.5 mm. This underscores the versatility of DL algorithms in identifying and alleviating noise, surpassing the PCA method in this instance. However, our study revealed no transferability beyond static phantom measurements; for in vivo measurements, as presented in our study, the current models are inapplicable considering the noise model used in training. The idealized noise types utilized during our training sessions did not appear to align with the complex noise environment encountered in in vivo experiments. For example, variables such as respiratory- or intestinal movement-related signal fluctuations in the abdomen degrade the quality of our neural models, thus leading to misinterpretations. In addressing potential enhancements to our noise reduction algorithms, it is vital to acknowledge that while our models showed promise, particularly with the in vitro data, they were trained on idealized noise models. To address the nuanced noise profiles of the in vivo data, it is essential to include more comprehensive noise models during training. Incorporating elements such as respiratory or intestinal movement-related signal fluctuations can result in algorithms that are more apt for real-world clinical scenarios. Therefore, future efforts to achieve flawless transferability to the in vivo data will require a comprehensive representation of noise and movement dynamics during the training phase, which might be a resource-intensive endeavor.

Moreover, even when trained with varying configurations and MR parameters, our models were constrained to a matrix of 128 × 128 pixels and 41 normalized offset frequencies. This limitation means that any deviation from these parameters requires a new cycle of data generation and re-training. Specifically, data generation was cumbersome as we utilized the framework developed by Zaiss et al., which operates on MATLAB (Matlab R2018a, Natick, MA, USA) [25,26] and cannot be parallelized without additional licenses. In our study, generating the 10,000 phantoms without multiprocessing on a standard clinical computer (CPU: Intel^®^ Xeon^®^ W-1250P @4.10 GHz with 16 GB RAM) took approximately 4000 h. The subsequent training in Python was performed on the clinic’s internal GPU server, concluding in just 3 h. It is notable that a recent tool by Herz et al. enables the computation of Z spectra based on C, potentially significantly accelerating the data generation [46]. This development offers potential for rapidly generating phantoms in future research, allowing flexibility in the resolution and dynamic count. In contrast, the PCA methods are readily adaptable and versatile. However, it is crucial to highlight that specific challenges emanate from the selected network architectures. In recent years, U-Net-based models have primarily shown efficacy in image evaluation [47], as corroborated by our study. However, alternative model architectures, like Long Short-Term Memory (LSTM) [48], have been particularly effective in the signal evaluation of speech and can handle data of variable lengths. The exploration of such methodologies is warranted in future studies for CEST imaging, allowing a trained model to analyze images with different numbers of dynamics. While the adaptability and precision of neural networks, such as ResUNet, are commendable, they come with trade-offs in terms of computational demands. Training these models, even at the specific resolution of 128 × 128 pixels used in our study, necessitates robust GPUs and extended computational hours. This underscores the need to strike a balance between algorithmic complexity and computational feasibility. To further the advancements in this field and build upon our study’s findings, we suggest several key areas for exploration. These encompass refined in vivo noise modeling, the investigation of diverse neural network architectures, ensuring greater model flexibility and streamlining the computational processes for efficiency.

Furthermore, transfer learning has established itself as a cornerstone in the field of medical image analysis [49,50], as it offers the possibility to use the performance of already existing trained models for new but similar tasks. Especially in scenarios where data availability is limited or where domain-specific fine-tuning is desired, transfer learning can significantly streamline the modelling process. In the context of our study, future studies could potentially use transfer learning to improve the adaptability of models generalized based on the simulated data, especially when moving from in vitro to in vivo scenarios.

Although the in silico generated data and NN training were successfully applied to the phantom data, it is important to highlight that artefacts were visible in the ROIs. Upon closer inspection, especially for the phantom with a layer thickness of 10 mm, a checkerboard-like pattern could be observed in the resulting MTR_asym_ values due to the convolutions. Although these patterns did not affect the average value, they could be a problem in detecting tiny defects, e.g., in cartilage. Even though such approaches can denoise 95% of the image more effectively than methods such as PCA, they are prone to local artifacts, which limits their clinical application. The PCA method proved to be robust in all experiments.

We have obtained promising results and comprehensive analyses in our study. Nevertheless, some limitations must also be considered:(1)Resolution and Offset Frequency Limitations: Our study’s results were anchored on a resolution of 128 × 128 pixels and 41 offset frequencies. As discussed, the number of offset frequencies is intricately linked to the number of feasible features for PCA. Notably, some studies, such as those cited [51,52], operated with fewer than 30 offset frequencies. This can potentially compromise the denoising performance, especially when utilizing the PCA method. For higher image resolutions, it is expected that the BM3D approach becomes applicable, as shown in previous studies [12], while NN training becomes more time-consuming and requires better GPUs.(2)Specific Dataset Limitations: Our evaluations were primarily centered on a phantom and an in vivo dataset that was limited to the lumbar IVDs. As discussed, other body regions might present unique artifacts, leading to varied noise distributions that our study did not account for.(3)Comparative Analysis Limitations: In our evaluations, the PCA method consistently outperformed both the NLM and BM3D techniques. However, it is pivotal to note that our comparative exploration was restricted to these specific denoising methods. As mentioned, other denoising techniques could potentially offer enhanced results. For instance, recent work by Chen et al. showcased a k-means clustering strategy designed to accelerate Lorentzian evaluations while inherently reducing noise [53]. Further, as discussed, methods such as the Image Downsampling Expedited Adaptive Least-squares (IDEAL) [7,54] have been proposed as effective alternatives for reducing noise during Lorentzian analyses.(4)Noise Model Disparities: As we emphasized in our discussion, the idealized noise models used during our training sessions seemed misaligned with the intricate noise landscapes of in vivo experiments, particularly due to variables like respiratory or intestinal movement-related signal fluctuations.(5)Potential for Local Artifacts: As discussed, even though strategies like neural networks can effectively denoise a significant portion of the image, they are susceptible to local artifacts, which can hinder their broader clinical applications.

## 5. Conclusions

Our exploration of noise reduction in CEST-MRI data using neural networks illuminates both the potentials and challenges of modern computational imaging. While neural networks show promise under specific conditions, conventional PCA methods consistently provide reliable results, notably outperforming NLM and BM3D.

The application of ResUNet stands out for its ability to detect subtle concentration differences. However, translating these findings from simulated to in vivo scenarios remains a central challenge. This underscores the necessity for designing future CEST-MRI studies with advanced noise management strategies.

In summary, our study paves the way for future research while emphasizing the need for continuous optimization and collaboration to fully harness the capabilities of advanced noise reduction techniques in CEST-MRI.

## Figures and Tables

**Figure 1 diagnostics-13-03326-f001:**
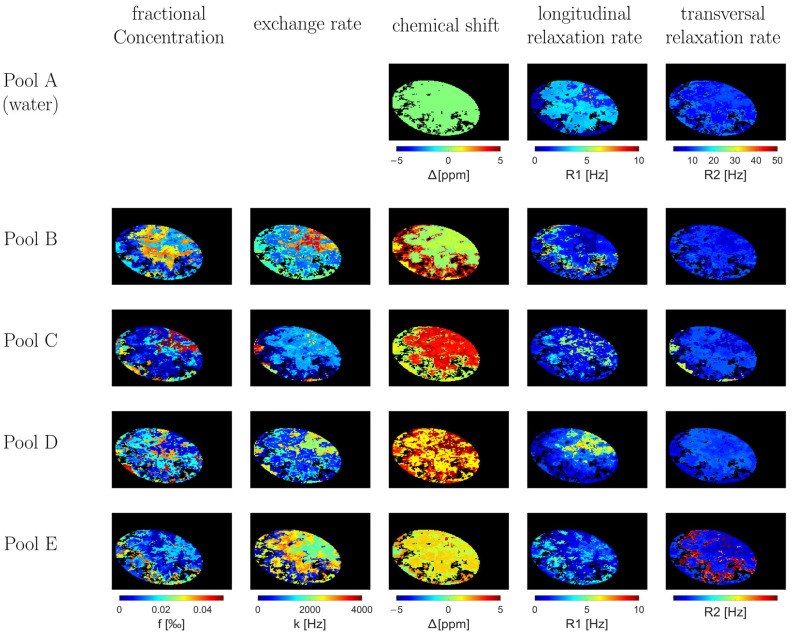
Synthesized In Silico Phantom Pool Parameter Maps: This illustration presents a systematically conceived in silico phantom generated via iterative layering of geometric forms and the corresponding spatial distribution of pool parameters. The CEST effects for each voxel are calculated using the Bloch–McConnell equations.

**Figure 2 diagnostics-13-03326-f002:**
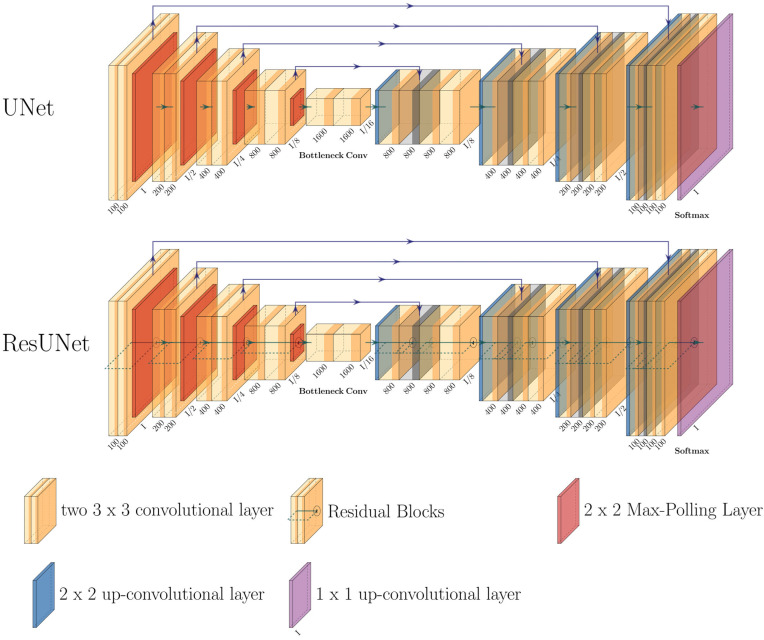
Comparative Architectures of UNet and ResUNet: This figure depicts schematic representations of the standard UNet and its advanced derivative, the ResUNet, both modified for CEST dataset denoising. Each subordinate layer clearly displays channel numbers juxtaposed with image magnitude relative to primary input dimension, denoted as “I”.

**Figure 3 diagnostics-13-03326-f003:**
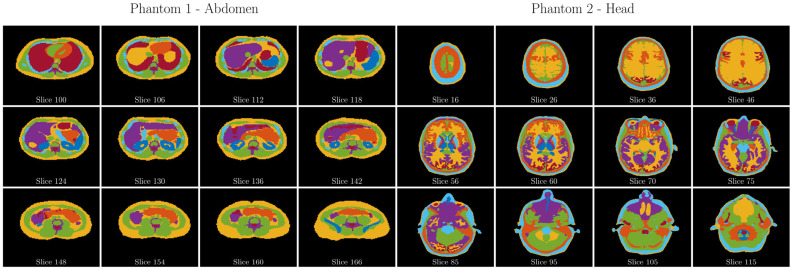
Visualization of Color-coded Layers in Zubal Phantoms: A visual representation of selected layers and corresponding color codings within Zubal phantoms is provided, depicting a subset of the overall dataset. Each unique color represents a specific region with a generated Z spectrum, based on parameters detailed in Table 1 and calculated using the Bloch–McConnell equations.

**Figure 4 diagnostics-13-03326-f004:**
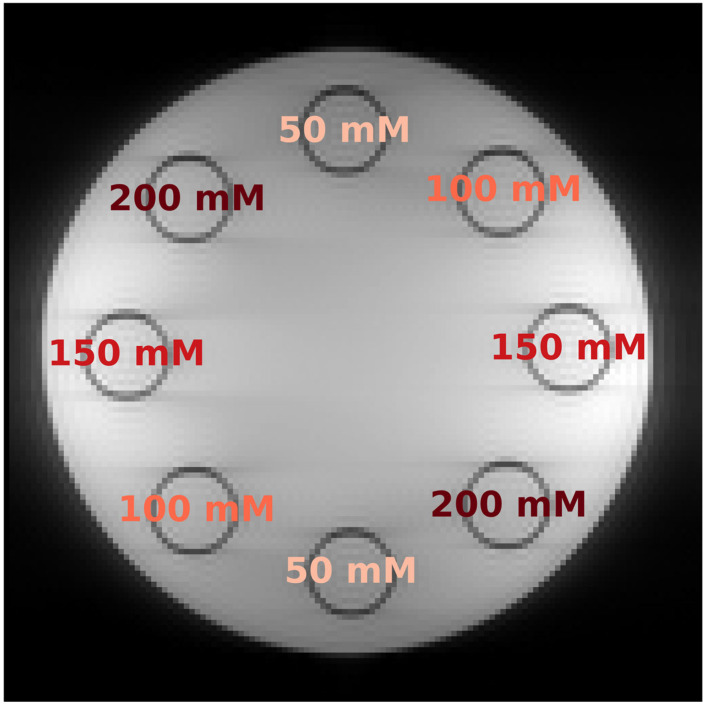
Creatine Concentration Distribution in Phantom: This figure displays an MR image of the 8-chamber phantom at 300 pm, illustrating the positions of sample tubes filled with varying concentrations of creatine.

**Figure 5 diagnostics-13-03326-f005:**
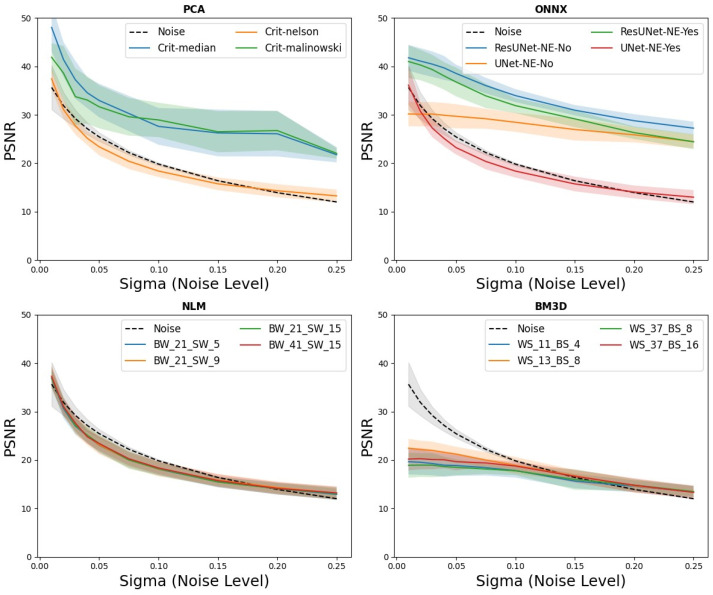
Comparative PSNR Results Across Various Noise Sigma Levels: This figure displays PSNR results for different noise levels and denoising methods determined over 200 in silico phantoms. It illustrates the efficiency of methods like PCA and ResUNets in noise reduction and the ineffectiveness of methods like BM3D in low-noise reduction conditions.

**Figure 6 diagnostics-13-03326-f006:**
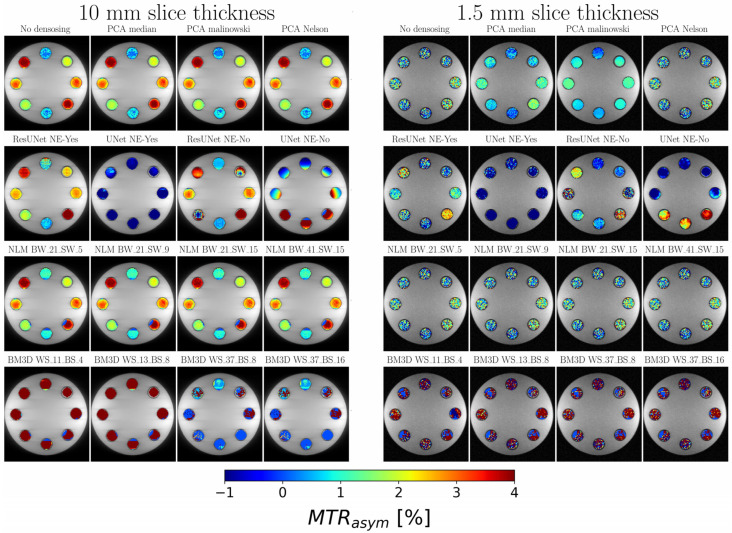
Analysis of MTR_asym_ Map Overlays and Noise Reduction for Different Slice Thicknesses and Methods: This figure visually presents MTR_asym_ map overlays from in vitro phantom studies, highlighting methodological annotations and resulting MTR_asym_ metrics for each tube, providing insights into the impact of different methodologies on noise reduction and MTR_asym_ values at varying slice thicknesses.

**Figure 7 diagnostics-13-03326-f007:**
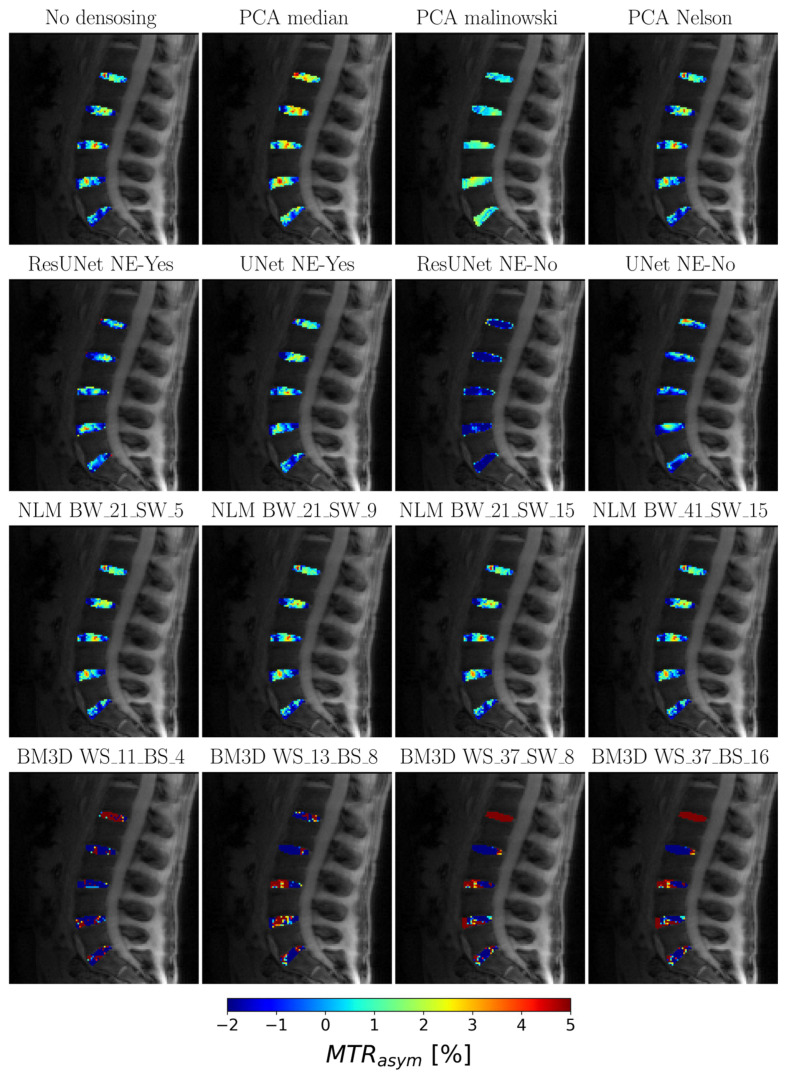
Overlay of MTR_asym_ Map on In Vivo IVD Measurement: The overlay of the MTR_asym_ map on the in vivo IVD measurement is depicted with methodological annotations, showcasing the variations in noise reduction and delineation between NP and AF using different methods such as PCA, neural approaches, and BM3D methods.

**Table 1 diagnostics-13-03326-t001:** Overview of the specified chemical exchange and sequence parameters for the CEST phantoms.

Exchange Parameters			Sequence Parameters	
Water (Pool 1)	T1 (s)	0.5–2.5	Δω (ppm)	4–6
	T2 (ms)	40–500	N	8–40
	Δ (ppm)	0	Tp (ms)	20–100
Metabolites (Pool 2–5)	T1 (s)	0.5–2.5	DC	0.5
	T2 (ms)	1–20	TE (ms)	2–40
	Exchange rate k (Hz)	50–4000	TR (ms)	11–60
	Fractional concentration (mM)	0–800	Dyn	50
	Δ (ppm)	0.5–5.0		

Abbreviations: T1—Longitudinal relaxation time, T2: Transverse relaxation time, Δω—Frequency acquisition range, N—Number of saturation pulses, Tp—Pulse duration, Δ—Chemical shift difference to water, DC—Duty Cycle, TE—Echo time, TR—Repetition time, Dyn—Number of equidistant offset frequencies in the Z spectra.

**Table 2 diagnostics-13-03326-t002:** MRI parameters for both CEST experiments.

	In Vitro CEST	In Vivo CEST
TE (ms)	5.76	3.50
TR (ms)	11 *	2500
Flip Angle (°)	10	15
Slices	1	1
Slice Thickness (mm)	10.0 and 1.5	6.0
FoV (mm × mm)	128 × 128	200 × 200
Pixel Size (mm × mm)	1.0 × 1.0	1.6 × 1.6
B1 (µT)	0.4	0.9
tp (ms)	50	100
td (ms)	50	100
DC	0.5	0.5
n	15	40
Duration (min:s)	3:37	12:05

Abbreviations: TE: Echo Time, TR: Repetition Time, FoV: Field of View, B1: Magnetic Field Strength, tp: Pulse Time, td: Delay Time, DC: Duty Cycle, and n: Number of saturation pulses, * break between the offsets 2 s.

## Data Availability

The code used for the generation of the data as well as for the analysis is freely available on GitHub (https://github.com/MPR-UKD/CEST-Generator, last access on: 7 May 2023, https://github.com/MPR-UKD/CEST-Denoise, last access on: 7 May 2023), and the raw data can also be made available by the authors upon reasonable request.

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
