# Peer review of "Deep Learning-Based Denoising of CEST MR Data: A Feasibility Study on Applying Synthetic Phantoms in Medical Imaging"

_diagnostics, 2023, doi:10.3390/diagnostics13213326_

Round 1
Reviewer 1 Report
Comments and Suggestions for Authors
1. What motivated the exploration of noise reduction methods for CEST MRI data, as highlighted in the introduction?
2. How does achieving a high signal-to-noise ratio (SNR) impact the detection of small CEST effects in MRI?
3. Can you explain how traditional metrics like Magnetization Transfer Ratio Asymmetry (MTRasym) and Lorentzian analyses are affected by image noise in CEST MRI?
4. Could you elaborate on the application of Bloch-McConnell equations in generating synthetic CEST images for this study?
5. How were autoencoders utilized in this study to denoise CEST images, and how were they compared to traditional denoising methods?
6. What were the specific datasets used to evaluate the performance of noise reduction algorithms, and how were they selected or prepared?
7. Can you provide more details on the ResUNet methods and how they were employed in noise detection and suppression?
8. What were the key findings regarding the performance of neural networks, particularly ResUNet, in noise detection and suppression for CEST images based on the synthetic phantoms?
9. How did the performance of neural networks compare to traditional denoising methods at various noise levels?
10. Were there any notable differences in noise detection and suppression when applying the denoising methods to in vitro versus in vivo data, and if so, what were they?
11. What are the implications of the observed differences in noise detection and suppression between neural network-based approaches and traditional denoising methods?
12. How might the effectiveness of noise reduction algorithms, especially neural networks, influence the quantitative estimation of biomolecule concentrations in tissues using CEST MRI?
13. In what ways could the findings of this study impact the future development and application of noise reduction techniques for CEST MRI in clinical practice or research?
14. Were there any limitations encountered in the application of neural networks to in vivo data, and how might these challenges be addressed in future research?
15. How does the use of synthetic phantoms affect the generalizability and applicability of the noise reduction algorithms to real-world clinical scenarios with human subjects?
16. Can you discuss the potential for further improvement or refinement of the noise reduction algorithms, particularly in addressing different noise distributions in in vivo data?
17. How might the findings of this study contribute to advancing the field of medical imaging and specifically benefit the analysis of CEST MRI data in clinical or research settings?
18. What are the future research directions or areas of investigation that could build upon the insights gained from this study?
19. Did the study identify any trade-offs or considerations regarding computational complexity or computational resources when implementing the noise reduction algorithms?
20. How might the knowledge gained from this study influence the design of future CEST MRI studies, particularly in terms of noise management and data quality?
Comments on the Quality of English LanguagePlease provide proof of English editing certificate.
Author Response
Title Deep learning based denoising of CEST MR data: A feasibility study on applying synthetic phantoms in medical imaging
Manuscript ID: diagnostics- 2665334
Journal: Diagnostics
Responses to the Reviewers’ Comments
The authors thank the reviewers for their careful revision of the manuscript and valuable comments, which are addressed and considered in the revised version of the manuscript. Please note that all changes made to the manuscript have been highlighted using the “Track changes”-mode in Microsoft Word and are detailed in this document, too. Please note that line numbers refer to the main text body of the revised document.
R1 Comment#1: “What motivated the exploration of noise reduction methods for CEST MRI data, as highlighted in the introduction?”
Authors’ Response: The motivation to explore noise reduction methods for CEST MRI data arises from its inherent challenges in detecting subtle CEST effects, especially when perturbed by noise. Given that MTRasym, a key metric in CEST, relies on comparing both sides of a Z-spectrum, any noise can skew its results, leading to inaccurate biomolecule concentration assessments. Thus, effective denoising is pivotal for the accuracy and reliability of CEST imaging outcomes.
Authors‘ Action: We have clarified this motivation in the introduction and added an explanation on the significance of denoising in the context of MTRasym and its mathematical representation in the introduction.
“Mathematically, MTRasym is defined as the difference between the signals at positive and negative frequency offsets with respect to the water resonance. This metric essentially compares the two sides of a Z-spectrum. The symmetry of the Z-spectrum is disturbed by the presence of noise, leading MTRasym to provide skewed results. Lorentzian analysis involves fitting the CEST spectrum to Lorentzian line shapes, noise in the spectrum can invalidate this assumption, especially since peaks are no longer symmetrical.” (line 54ff)
R1 Comment#2: “How does achieving a high signal-to-noise ratio (SNR) impact the detection of small CEST effects in MRI?”
Authors’ Response: Achieving a high SNR is paramount in CEST MRI. A high SNR ensures that the desired CEST effects are distinct from background noise, facilitating accurate detection and interpretation. This is particularly crucial when using metrics like MTRasym, as noise can disrupt the symmetry of the Z-spectrum, leading to skewed values and potentially inaccurate biomolecular concentration assessments.
Authors‘ Action: In light of this valid comment, we have added further clarity on the significance of a high SNR in the introduction:
“Achieving a high SNR ensures that the CEST effects, no matter how subtle, are distinctly discernible against the background noise. This becomes especially crucial when working with low concentrations or when the effects are inherently small, as a compromised SNR could lead to potential misinterpretations or even missed detections.” (line 47ff)
R1 Comment#3: “Can you explain how traditional metrics like Magnetization Transfer Ratio Asymmetry (MTRasym) and Lorentzian analyses are affected by image noise in CEST MRI?”
Authors’ Response: We appreciate the reviewer's inquiry. As detailed in our response to Comment#1, noise can significantly impact the MTRasym due to its reliance on the symmetry of the Z-spectrum. Any noise can skew this symmetry, leading to inaccurate MTRasym values. Similarly, for Lorentzian analyses, noise can distort the expected Lorentzian line shapes in the CEST spectrum, leading to potential inaccuracies in the derived parameters.
R1 Comment#4: “Could you elaborate on the application of Bloch-McConnell equations in generating synthetic CEST images for this study?”
Authors’ Response: We thank the reviewer for this important question since it is crucial part of our study. The Bloch-McConnell equations are differential equations that delineate the behavior of nuclear magnetization in a multi-pool exchange environment subjected to radiofrequency irradiation. For CEST MRI, these equations facilitate an accurate depiction of magnetization transfer dynamics between solute and solvent pools. This ability to mathematically model exchange processes enables the generation of synthetic images that closely emulate real CEST MRI dynamics.
Authors‘ Action: We have incorporated a more detailed description emphasizing the role of the Bloch-McConnell equations in our synthetic image generation process within the section "2.2. Generation of in silico phantoms"
“The foundation of our synthetic image generation lies in the Bloch-McConnell equations, a set of differential equations that describe the evolution of nuclear magnetization in a multi-pool exchange system under the influence of radiofrequency irradiation. In the context of CEST MRI, these equations allow for an accurate representation of the magnetization transfer between the solute and solvent pools. The capacity to simulate these exchange dynamics provides a nuanced understanding of CEST contrast mechanisms, making it possible to generate realistic synthetic images that replicate the intricacies of actual CEST MRI scenarios.” (line 124ff)
R1 Comment#5: “How were autoencoders utilized in this study to denoise CEST images, and how were they compared to traditional denoising methods?”
Authors’ Response: Autoencoders are neural network architectures designed for unsupervised learning of efficient codings. In the context of our study, autoencoders were employed to denoise CEST images by first compressing the noisy image into a lower-dimensional latent space and then reconstructing a denoised version from this compressed representation. The effectiveness of autoencoders in denoising was then quantitatively evaluated using the peak signal-to-noise ratio (PSNR) as the primary assessment criterion. Specifically, for our digital phantoms, the PSNR provided an accurate measure of the fidelity of the reconstructed signal profiles throughout each Z-spectrum. This quantitative metric allowed us to directly compare the performance of autoencoders against traditional denoising methods in an unbiased manner. For MRI data, the evaluation also included a visual inspection of the generated MTRasym maps to ensure the quality and consistency of the denoised images.
Authors‘ Action: We have added an explanation on the application of autoencoders in our study and their comparison to traditional denoising methods in the section "2.8 Evaluation metrics":
“2.8 Evaluation metrics
The investigation of the trained autoencoders regarding their ability to denoise CEST images by compressing and reconstructing the image data was the focus of our study. Analogous to the analytical methods, a noisy Z-spectrum is passed, and a denoised Z-spectrum is returned. To provide a reliable quantitative measure for evaluating the effectiveness of the various denoising strategies, the peak signal-to-noise ratio (PSNR) was used as the primary assessment criterion. […]” (line 356ff)
R1 Comment#6: “What were the specific datasets used to evaluate the performance of noise reduction algorithms, and how were they selected or prepared?”
Authors’ Response: For our study's comprehensive evaluation, we employed a mix of synthetically generated, in vitro, and in vivo datasets:(1) Synthetically Generated Phantoms: These were generated using in silico techniques. The 10,000 random 2D CEST datasets created were used primarily for training purposes, ensuring no typical in vivo patterns were present, and ensuring unbiased model training. (2) Zubal Phantom: This standardized phantom provided a consistent basis for our evaluations. (3) In Vitro Data: We used an MR-compatible phantom with slots for eight test tubes containing varying concentrations of creatine. Post-preparation, we took two separate recordings of the phantom using the CEST sequence, with one acting as a "ground truth" and the other introducing a higher noise profile for evaluation. (4) In Vivo Data: CEST MRI examinations were performed on intervertebral discs (IVDs) following the protocol described in section 2.7, "MRI validation". These examinations were pivotal for validating the transferability of our models from synthetic and controlled environments to real-world clinical scenarios. The datasets were meticulously chosen to provide a thorough evaluation of the denoising methods, ranging from controlled synthetic simulations to real-world in vivo conditions. The overarching theme of our study, as highlighted in the title, centers on the transferability of these denoising methods, thus necessitating such diverse datasets.
R1 Comment#7: “Can you provide more details on the ResUNet methods and how they were employed in noise detection and suppression?”
Authors’ Response: Certainly. The UNet and its evolved counterpart, ResUNet, are deep convolutional neural network architectures that are tailored for medical image processing tasks, notably in image denoising. Their distinct architecture, capable of conserving spatial details across encoding and decoding stages, makes them intrinsically proficient for denoising. The ResUNet stands out with its augmented design containing extra residual blocks, which ensure an optimized flow of data throughout the network. This results in scrupulous noise reduction while retaining the pivotal image features. In our study, we leveraged these architectures to address the challenges of noise in CEST-MRI data. We devised a two-pronged strategy for each architecture: One model aimed at noise estimation and its subsequent subtraction from the image, and another geared towards direct noise removal from the image data, ensuring a comprehensive noise suppression approach.
Authors‘ Action: We have augmented the "2.3. Neural Network Architectural Design" section with a description underscoring the inherent capabilities of the UNet and ResUNet architectures in the context of our study:
“The UNet and its modification, ResUNet, are deep convolutional neural network architectures that have shown significant prowess in medical image processing tasks, especially in image denoising. Their design, which conserves spatial context throughout the encoding and decoding phases, makes them inherently adept for denoising applications. The ResUNet, with its enhanced structure featuring additional residual blocks, facilitates an efficient flow of information through the network. This ensures a meticulous suppression of noise while preserving the essential features of the image [29].” (line 166ff)
R1 Comment#8: “What were the key findings regarding the performance of neural networks, particularly ResUNet, in noise detection and suppression for CEST images based on the synthetic phantoms?”
Authors’ Response: Our study found that ResUNet models, when trained on in-silico-generated phantom data, were particularly adept at denoising CEST MRI images. Specifically, ResUNet models consistently demonstrated superior noise suppression capabilities, especially for images with substantial noise resulting from thermal processes. Among the two distinct ResUNet methods employed, the one that first identified the noise and subsequently combined it with the original image (termed ResUNet-NE-yes) slightly outperformed the alternative approach that directly reconstructed the denoised image. These findings underscore the potential of neural networks, especially ResUNet, as robust denoising tools for CEST MRI, and they further emphasize the utility of synthetic phantoms for refining such neural methods.
Authors‘ Action: To elucidate our findings, we have incorporated an explicit discussion on the performance of ResUNet in the context of our synthetic phantoms in the “Discussion” section:
“The performance assessment of ResUNet on our synthetic phantoms revealed its significant capabilities in noise suppression, particularly for thermal process-related noise. Among the evaluated neural architectures, ResUNet-NE-yes, which employs a two-step approach of noise identification followed by image combination, showcased marginally superior performance over the direct denoising method. These outcomes reiterate the potential of neural networks, especially architectures like ResUNet, in enhancing CEST MRI images, and further emphasize the instrumental role of in-silico phantoms in refining and validating such neural methods.” (line 455ff)
R1 Comment#9: “How did the performance of neural networks compare to traditional denoising methods at various noise levels?”
Authors’ Response: In our digital phantom (in silico) analysis, we systematically evaluated the denoising performance of neural networks against traditional methods across varying noise levels. Our results, as depicted in Figure 5, indicate that neural networks, particularly the models we trained, consistently outperformed traditional methods like PCA, especially at higher noise intensities. While PCA showed a steady increase in PSNR values across noise levels, the performance exhibited significant variance across different layers of the phantom. On the other hand, our neural networks maintained commendable denoising capabilities throughout, achieving superior PSNR values in the range of 30-40, particularly prominent above a noise intensity of 0.05.
Authors‘ Action: We have added detailed insights into the comparative performance of neural networks and traditional denoising methods based on varying noise levels in the section "3.2 Digital phantom (in silico) analysis."
“The simulations showed a clear relationship between denoising performance, the magnitude of the noise perturbation, and the denoising strategy used (Figure 5). When applying the PCA algorithm, both the “median” and “Malinowski” criteria led to a steady increase in PSNR values through the varying noise levels. This trend suggests that as noise levels rose, the PCA methods attempted to retain more signal information. However, there was significant variance between the different layers of the phantom, which was reflected in substantial standard deviations. In contrast, the Nelson criterion provided PSNR results comparable to the noise, indicating excessive component retention, implying potential overfitting in noise representation. Although the BM3D and NLM techniques also produced successful results as the noise level increased, they proved in-applicable in certain scenarios, potentially due to their inherent design for more generic noise patterns. Notably, the NN models, which were effective during the training phase, continued to show commendable denoising capabilities, achieving PSNR ranges of 30 - 40. This highlights the adaptability and robustness of neural networks in handling diverse noise structures. Their performance distinctly outperformed the PCA-based methods, which was particularly evident above a noise intensity of 0.05, underscoring the superiority of trained neural models in challenging noise conditions.” (line 385ff)
R1 Comment#10: Were there any notable differences in noise detection and suppression when applying the denoising methods to in vitro versus in vivo data, and if so, what were they?”
Authors’ Response: Yes, there were discernible differences when applying the denoising methods to in vitro versus in vivo data. The in vitro experiments were largely consistent, owing to the controlled nature of the samples. However, the in vivo data, due to its inherent complexity and diversity of biological tissues, showed varied results. The neural networks, while effective on in vitro and in silico datasets, showed differences in performance on in vivo data due to unanticipated noise distributions and the presence of factors like motion artifacts from respiration and peristalsis. These distinctions and challenges, especially in the context of ResUNet's performance and the role of in silico phantoms, are comprehensively elaborated upon in the discussion section of our paper (Section 4: Discussion).
“However, a significant difference was seen when comparing in silico and in vitro datasets with in vivo data. This divergence is primarily due to different noise distributions, which were not considered during the NN training phase but appeared in vivo. Visible movements in the image space due to respiration and peristalsis, which were not visible in the ROIs, but the NNs analyzed the whole image without a prior selection of ROIs.” (line 468ff)
R1 Comment#11: “What are the implications of the observed differences in noise detection and suppression between neural network-based approaches and traditional denoising methods?”
Authors’ Response: The observed differences between neural network-based approaches and traditional denoising methods underscore the advancements in modern computational methods and their potential in enhancing medical imaging processes. Specifically, neural network-based approaches, such as the ResUNet utilized in our study, demonstrated superior capabilities, especially in scenarios with significant noise perturbations. This suggests that such architectures can offer more robust noise suppression, particularly when trained with comprehensive data that captures a wide range of noise distributions. Traditional denoising methods, while effective to a certain extent, may not be as adaptive or efficient in handling complex noise structures inherent in CEST MRI, as revealed by our study. The flexibility and adaptability of neural networks, informed by extensive training data, enable them to discern and address intricate noise patterns that might be challenging for conventional methods. The implications of these findings are manifold. First, they pave the way for the broader adoption of neural network-based denoising methods in CEST imaging, which could lead to improved diagnostic accuracy and reliability. Furthermore, the superior performance of these networks could reduce the need for repeated scans, potentially decreasing patient discomfort and exposure. Lastly, these results highlight the importance of continued research and development in the realm of deep learning for medical imaging to harness the full potential of these advanced computational tools.
R1 Comment#12: “How might the effectiveness of noise reduction algorithms, especially neural networks, influence the quantitative estimation of biomolecule concentrations in tissues using CEST MRI?”
Authors’ Response: The effective suppression and detection of noise, especially with neural network-based techniques, profoundly influence the quantitative estimation of biomolecule concentrations in tissues when employing CEST MRI. One crucial point to note is that the determination of concentrations often rests on the differentiation of MTRasym values, which are ascertained at varying B1 field strengths. Given the sensitivity of these measurements to noise, it is paramount that data remains consistent and devoid of noise artifacts across multiple scans.
Enhanced denoising capabilities, particularly from neural networks, can provide greater clarity and accuracy in images. This clarity directly translates into more reliable estimations of biomolecular concentrations. We have expounded on this correlation in our discussion section, emphasizing the notable merits of neural network techniques. These methods not only allow for superior noise suppression but also offer adaptability in catering to diverse noise patterns, thereby enabling more precise measurements of biomolecular concentrations. The resultant improved diagnostic and prognostic potential of CEST MRI further underscores its value in clinical studies.
Authors‘ Action: We have integrated a discussion on the implications of these observed differences in the "4. Discussion" section.
“Furthermore, the proficiency of noise reduction algorithms, especially when utilizing neural network-based strategies, plays a pivotal role in the quantitative estimation of biomolecule concentrations in tissues via CEST MRI. As concentration determinations typically depend on discerning differences in MTRasym values obtained at varied B1 field strengths, it becomes essential that the data remains consistent and free from noise-induced distortions across multiple measurements. With the enhanced image clarity offered by neural network denoising, we can ensure a more dependable extraction of these values, leading to accurate biomolecular concentration measurements. This advancement, in turn, magnifies the diagnostic and prognostic acumen of CEST MRI, solidifying its stature as a trusted tool in clinical studies.” (line 474ff)
R1 Comment#13: “ In what ways could the findings of this study impact the future development and application of noise reduction techniques for CEST MRI in clinical practice or research?”
Authors’ Response: The findings of our study highlight the potential advantages of utilizing neural network-based approaches, particularly the ResUNet architecture, for denoising CEST MRI images. In clinical practice, enhanced image clarity can significantly improve the diagnostic precision, enabling earlier and more accurate disease detection and monitoring. For research, these advancements pave the way for more intricate studies that require high-resolution and noise-free images, potentially opening new avenues for CEST MRI applications. As these neural network methods continue to mature, they may become a standard component of CEST MRI processing pipelines, driving both clinical and research advancements in the field.
Authors‘ Action: We have discussed the potential implications and broader impacts of our findings in the "4. Discussion" section, emphasizing the potential transformative role of neural network-based denoising techniques in both clinical and research settings.
“These outcomes reiterate the potential of neural networks, especially architectures like ResUNet, in enhancing CEST MRI images, and further emphasize the instrumental role of in silico phantoms in refining and validating such neural methods.” (line 462ff)
R1 Comment#14: “Were there any limitations encountered in the application of neural networks to in vivo data, and how might these challenges be addressed in future research?”
Authors’ Response: Indeed, our investigation delineated certain limitations associated with the deployment of neural networks to in vivo datasets. In our revised manuscript, we have methodically restructured the "Limitations" subsection within the "4. Discussion" segment to provide a more comprehensive elucidation of these challenges. A salient concern was the discord between the idealized noise models utilized during the training phase and the multifaceted noise characteristics inherent in in vivo scenarios. This dissonance was further compounded by factors such as respiratory or intestinal motion-induced signal perturbations. Future endeavors will necessitate methodological refinements to our neural models to better accommodate and adapt to these intricate clinical realities.
Authors‘ Action: Enhanced and restructured the "Limitations" subsection in the "4. Discussion" section for improved clarity and depth.
“We have obtained promising results and comprehensive analyses in our study. Nevertheless, some limitations must also be considered:
(1) Resolution and Offset Frequency Limitations: Our study's results were anchored on a resolution of 128 x 128 pixels and 41 offset frequencies. As discussed, the number of offset frequencies is intricately linked to the number of feasible features for PCA. Notably, some studies, such as those cited [49,50], operated with fewer than 30 offset frequencies. This can potentially compromise the denoising performance, especially when utilizing the PCA method. For higher image resolutions, it is expected that the BM3D approach becomes applicable, as shown in previous studies [12], while NN training becomes more time-consuming and requires better GPUs.
(2) Specific Dataset Limitations: Our evaluations were primarily centered on a phantom and an in vivo dataset that was limited to the lumbar IVDs. As discussed, other body regions might present unique artifacts, leading to varied noise distributions that our study did not account for.
(3) Comparative Analysis Limitations: In our evaluations, the PCA method consistently outperformed both the NLM and BM3D techniques. However, it's pivotal to note that our comparative exploration was restricted to these specific denoising methods. As mentioned, other denoising techniques could potentially offer enhanced results. For instance, recent work by Chen et al. showcased a k-means clustering strategy designed to accelerate Lorentzian evaluations while inherently reducing noise [51]. Further, as discussed, methods such as the Image Downsampling Expedited Adaptive Least-squares (IDEAL) [7,52] have been proposed as effective alternatives for reducing noise during Lorentzian analyses.
(4) Noise Model Disparities: As we emphasized in our discussion, the idealized noise models used during our training sessions seemed misaligned with the intricate noise landscapes of in vivo experiments, particularly due to variables like respiratory or intestinal movement-related signal fluctuations.
(5) Potential for Local Artifacts: As discussed, even though strategies like neural networks can effectively denoise a significant portion of the image, they are susceptible to local artifacts, which can hinder their broader clinical applications.” (line 605ff)
R1 Comment#15: “How does the use of synthetic phantoms affect the generalizability and applicability of the noise reduction algorithms to real-world clinical scenarios with human subjects?”
Authors’ Response: The use of synthetic phantoms offers a controlled environment to develop and fine-tune noise reduction algorithms. It ensures consistent, reproducible results and allows for rigorous testing across various noise scenarios. However, while synthetic phantoms are instrumental in the initial development stages, the inherent differences between phantom and real-world clinical data may pose challenges. For instance, in vivo data often comes with complex noise profiles, including physiological motion artifacts that aren't present in synthetic phantoms. Hence, while synthetic phantoms provide an excellent starting point, it is crucial to validate and further refine algorithms using real-world clinical datasets to ensure generalizability and clinical applicability.
Authors‘ Action: We have emphasized the role of synthetic phantoms in the initial development and the need for real-world validation in the "4. Discussion" section to address concerns about the generalizability and applicability of our findings to clinical scenarios.
“In our study, we were able to demonstrate the potential of NNs trained only on simulated data and the transfer to in vitro data but also observed challenges for in vivo data. The idealized noise types utilized during our training sessions did not appear to align with the complex noise environment encountered in in vivo experiments. For example, variables such as respiratory- or intestinal movement-related signal fluctuations in the abdomen degrade the quality of our neural models, thus leading to misinterpretations.” (line 531ff)
R1 Comment#16: “ Can you discuss the potential for further improvement or refinement of the noise reduction algorithms, particularly in addressing different noise distributions in in vivo data?”
Authors’ Response: Thank you for highlighting this point. While our current models showed potential, they were primarily trained on idealized noise models. In vivo data, however, present a more intricate noise landscape, influenced by factors like respiratory or intestinal movement-related signal fluctuations. Future refinements should encompass more comprehensive noise models during training that incorporate these physiological intricacies and other potential sources of interference.
Authors‘ Action: We have expanded upon this aspect in the "Discussion" section, emphasizing the need for more complex training datasets that capture the diversity of noise distributions encountered in in vivo settings
“In addressing potential enhancements to our noise reduction algorithms, it's vital to acknowledge that while our models showed promise, particularly with in vitro data, they were trained on idealized noise models. To address the nuanced noise profiles of in vivo data, it's essential to include more comprehensive noise models during training. Incorporating elements such as respiratory or intestinal movement-related signal fluctuations can result in algorithms that are more apt for real-world clinical scenarios.” (line 550ff)
R1 Comment#17: “How might the findings of this study contribute to advancing the field of medical imaging and specifically benefit the analysis of CEST MRI data in clinical or research settings?”
Authors’ Response: The findings of our study highlight the pivotal role of advanced noise reduction techniques in the realm of CEST MRI. We delineate the nuanced potentials and challenges associated with the application of neural networks in contrast to traditional PCA methods. By improving denoising, we can significantly elevate the precision of CEST MRI images, thereby strengthening the reliability of biomolecular concentration estimations in tissues. Such advancements hold the promise of fostering more accurate diagnostic and prognostic evaluations, which can profoundly impact both clinical and research settings.
Authors‘ Action: We have extensively restructured and refined the "Conclusion" section to ensure that the implications of our findings for the broader field of medical imaging, particularly CEST MRI, are highlighted. The revised section encapsulates our contributions and their potential impact on clinical and research applications.
“5. Conclusions
Our exploration of noise reduction in CEST MRI data using neural networks illuminates both the potentials and challenges of modern computational imaging. While neural networks show promise under specific conditions, conventional PCA methods consistently provide reliable results, notably outperforming NLM and BM3D.
The application of ResUNet stands out for its ability to detect subtle concentration differences. However, translating these findings from simulated to in vivo scenarios remains a central challenge. This underscores the necessity for designing future CEST MRI studies with advanced noise management strategies.
In summary, our study paves the way for future research while emphasizing the need for continuous optimization and collaboration to fully harness the capabilities of advanced noise reduction techniques in CEST MRI.” (line 654ff)
R1 Comment#18: “ What are the future research directions or areas of investigation that could build upon the insights gained from this study?”
Authors’ Response: Our study has shed light on several facets of noise reduction in CEST MRI data, unveiling both the strengths and challenges of applying neural network methodologies. In light of our findings, we propose several pivotal areas for future exploration: (1) In Vivo Noise Modeling: Given the discrepancies observed between our idealized noise models and the intricate in vivo noise landscapes, a focused effort on creating more representative noise models would be crucial. (2) Exploration of Neural Network Architectures: While our study championed the capabilities of ResUNet, it is worth investigating other architectures like LSTM, which have demonstrated effectiveness in other domains such as signal evaluation. (3) Model Flexibility: To enhance the broad applicability of our models, research efforts can aim to make them adaptive to a wider range of resolutions and offset frequencies. (4) Computational Efficiency: Given the time-intensive nature of our data generation process, embracing tools that can expedite this, like the tool by Herz et al., (doi:10.1002/mrm.28825) would be invaluable.
Authors‘ Action: We have expanded upon the future research directions in the "Discussion" section, ensuring that readers and subsequent researchers are informed of the next logical steps in this research domain.
“To further the advancements in this field and build upon our study's findings, we suggest several key areas for exploration. These encompass refined in vivo noise modeling, the investigation of diverse neural network architectures, ensuring greater model flexibility, and streamlining computational processes for efficiency.” (line 584ff)
R1 Comment#19: “ Did the study identify any trade-offs or considerations regarding computational complexity or computational resources when implementing the noise reduction algorithms?”
Authors’ Response: Yes, we identified trade-offs, especially in the context of neural network training. While neural networks, particularly ResUNet, showcased adaptability and precision, they demand substantial computational resources. For instance, our models, even when constrained to a 128 x 128-pixel matrix, required advanced GPUs and extended training times. In contrast, while methods like PCA proved adaptable and versatile, neural network training, especially at higher resolutions, becomes more resource-intensive.
Authors‘ Action: The "Discussion" section has been augmented to reflect these considerations regarding computational complexity and resource demands.
“While the adaptability and precision of neural networks, such as ResUNet, are commendable, they come with trade-offs in terms of computational demands. Training these models, even at the specific resolution of 128 x 128 pixels used in our study, necessitates robust GPUs and extended computational hours. This underscores the need to strike a balance between algorithmic complexity and computational feasibility.” (line 579ff)
R1 Comment#20: “ How might the knowledge gained from this study influence the design of future CEST MRI studies, particularly in terms of noise management and data quality?”
Authors’ Response: Our study illuminates the pivotal role of effective noise reduction in enhancing data quality for CEST MRI. As we move forward, future CEST MRI studies can prioritize incorporating advanced denoising techniques, such as the ones explored in our research, right from the design phase. Emphasizing noise management will ensure that the data collected is of the highest quality, thereby ensuring more accurate and reliable results in both clinical and research settings.
Authors‘ Action: We have incorporated this perspective into the "Conclusion" section, emphasizing the importance of noise management in the design of future CEST MRI studies.
“5. Conclusions
Our exploration of noise reduction in CEST MRI data using neural networks illuminates both the potentials and challenges of modern computational imaging. While neural networks show promise under specific conditions, conventional PCA methods consistently provide reliable results, notably outperforming NLM and BM3D.
The application of ResUNet stands out for its ability to detect subtle concentration differences. However, translating these findings from simulated to in vivo scenarios remains a central challenge. This underscores the necessity for designing future CEST-MRI studies with advanced noise management strategies.
In summary, our study paves the way for future research while emphasizing the need for continuous optimization and collaboration to fully harness the capabilities of advanced noise reduction techniques in CEST MRI.” (line 654ff)
Reviewer 2 Report
Comments and Suggestions for Authors
This paper is well organized and technically sound. Abundant experiments have been implemented to reveal the performance.
Transfer learning is a popular way to generate image representations, and has been used in medical image analysis. Please discuss the methods in 'NAGNN: classification of COVID‐19 based on neighboring aware representation from deep graph neural network', and 'Detection of abnormal brain in MRI via improved AlexNet and ELM optimized by chaotic bat algorithm'.
The overall novelty is weak.
Please share your codes to the community such as GitHub so that people can follow your research.
It's always good practice to acknowledge the limitations of the study. This could include limitations in the data, the model, or the experimental setup. Acknowledging limitations shows deep understanding of the research and provides avenues for future work.
Author Response
Title Deep learning based denoising of CEST MR data: A feasibility study on applying synthetic phantoms in medical imaging
Manuscript ID: diagnostics- 2665334
Journal: Diagnostics
Responses to the Reviewers’ Comments
The authors thank the reviewers for their careful revision of the manuscript and valuable comments, which are addressed and considered in the revised version of the manuscript. Please note that all changes made to the manuscript have been highlighted using the “Track changes”-mode in Microsoft Word and are detailed in this document, too. Please note that line numbers refer to the main text body of the revised document.
R2 General Comment: “This paper is well organized and technically sound. Abundant experiments have been implemented to reveal the performance.”
Authors’ Response: First and foremost, we would like to express our heartfelt gratitude for your positive feedback and for recognizing the efforts we have invested in conducting comprehensive experiments. We truly appreciate your acknowledgment of the organization and technical soundness of our manuscript. Your constructive feedback not only inspires us but also aids in refining our research. In the following sections, we have addressed your comments point by point to ensure clarity and further enhance the manuscript.
R2 Comment#2: “Transfer learning is a popular way to generate image representations, and has been used in medical image analysis. Please discuss the methods in 'NAGNN: classification of COVID‐19 based on neighboring aware representation from deep graph neural network', and 'Detection of abnormal brain in MRI via improved AlexNet and ELM optimized by chaotic bat algorithm'.”
Authors’ Response: Thank you for emphasizing the significance of transfer learning in the realm of medical image analysis. Transfer learning has indeed emerged as a pivotal tool, allowing the leveraging of pre-trained models on new, yet similar, tasks. While the studies you mentioned present intriguing methodologies tailored to their specific objectives, our study's primary challenge revolved around denoising CEST MRI data. Pre-existing models that leverage transfer learning in noise reduction usually cater to 2D image data or, within the medical spectrum, 3D data. CEST imaging, on the other hand, necessitates dealing with signals across diverse offset frequencies. As our analytical methods have indicated, a holistic response over these frequencies is paramount — a sentiment echoed by the superior performance of PCA compared to NLM and BM3D. We opted for a more generalized model approach, rather than heavily relying on transfer learning from the simulated data. That said, the prospects of integrating transfer learning in subsequent CEST MRI explorations are undeniable and hold significant promise. Our findings distinctly illustrate that while generalization to other simulated phantoms and real phantoms was achieved, the same could not be said for in vivo data. In future endeavors, transfer learning might be the linchpin in minimizing the requisite data for fine-tuning our model to in vivo scenarios.
Authors‘ Action: In response to the reviewer's insightful feedback, we have integrated a section on transfer learning within the "Discussion" segment. Here, we reference the studies alluded to and expound on the potential of transfer learning as a subsequent step in our research journey.
“Furthermore, transfer learning has established itself as a cornerstone in the field of medical image analysis [49,50], as it offers the possibility to use the performance of already existing trained models for new but similar tasks. Especially in scenarios where data availability is limited or where domain-specific fine-tuning is desired, transfer learning can significantly streamline the modelling process. In the context of our study, future studies could potentially use transfer learning to improve the adaptability of models generalised based on simulated data, especially when moving from in vitro to in vivo scenarios.” (line 588ff)
R2 Comment#3: “Please share your codes to the community such as GitHub so that people can follow your research.”
Authors’ Response: We wholeheartedly agree with the reviewer's emphasis on open-source sharing to foster community collaboration and research reproducibility. In line with this, we have already made our 2D data generator, the deep learning model, and analytical methods available to the community via GitHub. Details and links to our GitHub repository are provided within the manuscript for ease of access by interested researchers.
“The digital framework we developed, available at [GitHub: https://github.com/MPR-UKD/CEST-Generator], ….” (line 141f)
“Furthermore, the final trained models, as well as the implemented denoising methods, can be used via a simple command line tool and are freely available on GitHub: https://github.com/MPR-UKD/CEST-Denoise.” (lines 380ff)
“Data Availability Statement: The code used for the generation of the data as well as for the analysis is freely available on GitHub (https://github.com/MPR-UKD/CEST-Generator, https://github.com/MPR-UKD/CEST-Denoise), and the raw data can also be made available by the authors upon reasonable request.” (line 693ff)
R2 Comment#4: “It's always good practice to acknowledge the limitations of the study. This could include limitations in the data, the model, or the experimental setup. Acknowledging limitations shows deep understanding of the research and provides avenues for future work.”
Authors’ Response and Action: We concur with the reviewer's assertion that acknowledging study limitations not only showcases a comprehensive grasp of the research but also paves the way for future explorations. We have always believed in the importance of this, and to ensure clarity and completeness, we have revised and elaborated upon the "Limitations" subsection at the end of our "Discussion" section.
“We have obtained promising results and comprehensive analyses in our study. Nevertheless, some limitations must also be considered:
(1) Resolution and Offset Frequency Limitations: Our study's results were anchored on a resolution of 128 x 128 pixels and 41 offset frequencies. As discussed, the number of offset frequencies is intricately linked to the number of feasible features for PCA. Notably, some studies, such as those cited [49,50], operated with fewer than 30 offset frequencies. This can potentially compromise the denoising performance, especially when utilizing the PCA method. For higher image resolutions, it is expected that the BM3D approach becomes applicable, as shown in previous studies [12], while NN training becomes more time-consuming and requires better GPUs.
(2) Specific Dataset Limitations: Our evaluations were primarily centered on a phantom and an in vivo dataset that was limited to the lumbar IVDs. As discussed, other body regions might present unique artifacts, leading to varied noise distributions that our study did not account for.
(3) Comparative Analysis Limitations: In our evaluations, the PCA method consistently outperformed both the NLM and BM3D techniques. However, it is pivotal to note that our comparative exploration was restricted to these specific denoising methods. As mentioned, other denoising techniques could potentially offer enhanced results. For instance, recent work by Chen et al. showcased a k-means clustering strategy designed to accelerate Lorentzian evaluations while inherently reducing noise [51]. Further, as discussed, methods such as the Image Downsampling Expedited Adaptive Least-squares (IDEAL) [7,52] have been proposed as effective alternatives for reducing noise during Lorentzian analyses.
(4) Noise Model Disparities: As we emphasized in our discussion, the idealized noise models used during our training sessions seemed misaligned with the intricate noise landscapes of in vivo experiments, particularly due to variables like respiratory or intestinal movement-related signal fluctuations.
(5) Potential for Local Artifacts: As discussed, even though strategies like neural networks can effectively denoise a significant portion of the image, they are susceptible to local artifacts, which can hinder their broader clinical applications.” (lines 605ff)
Reviewer 3 Report
Comments and Suggestions for Authors
Dear authors,
the article is original, and very well written. However, I have some suggestions which could improve the manuscript quality. Before that, I have to commend you for the excellent dataset description (nicely illustrated), the excellent sketches of the UNet and ResUNet.
1. Add most notable results at the end of the abstract,
2. In the introduction section, at the end, write the hypotheses in bullet form. Write the scientific contribution of this investigation. Write the outline of this paper by shortly describing what si written in the following sections.
3. In section 2 enumerate the equations. Lines 196,209,215, 227
4. In Figure 5 add grids to the plots.
5. First rename the conclusion section to conclusions since there is and should be more than one conclusion based on your extensive research. Second, the Conclusions section should be extended in the following way:
a) first paragraph of the conclusions section should describe shortly what was done in this research.
b) The second paragraph should provide the answers to the hypotheses in your research defined in the introduction section and based on the comments given in the discussion section.
c) The third paragraph should describe pros and cons of your research.
d) The fourth paragraph should contain potential directions for future research.
Author Response
Title Deep learning based denoising of CEST MR data: A feasibility study on applying synthetic phantoms in medical imaging
Manuscript ID: diagnostics- 2665334
Journal: Diagnostics
Responses to the Reviewers’ Comments
The authors thank the reviewers for their careful revision of the manuscript and valuable comments, which are addressed and considered in the revised version of the manuscript. Please note that all changes made to the manuscript have been highlighted using the “Track changes”-mode in Microsoft Word and are detailed in this document, too. Please note that line numbers refer to the main text body of the revised document.
R2 General Comment: Dear authors, the article is original, and very well written. However, I have some suggestions which could improve the manuscript quality. Before that, I have to commend you for the excellent dataset description (nicely illustrated), the excellent sketches of the UNet and ResUNet.”
Authors’ Response: We sincerely appreciate your positive feedback regarding the originality and structure of our manuscript. It is heartening to learn that our efforts to provide a detailed dataset description and our illustrations of the UNet and ResUNet were well-received. Your commendations provide us with encouragement to continually strive for clarity and comprehensiveness in our work. We are eager to address the suggestions you have provided, as we believe they will further enhance the quality of our manuscript. In the subsequent sections, we have responded to each of your comments point by point.
R3 Comment#1: “Add most notable results at the end of the abstract”
Authors’ Response: We appreciate the reviewer's astute observation and concur that the presentation of the most significant results within the abstract was insufficient.
Authors‘ Action: We have improved the abstract to the following version:
“Abstract: Chemical Exchange Saturation Transfer (CEST) magnetic resonance imaging (MRI) provides a novel method for analyzing biomolecule concentrations in tissues without exogenous contrast agents. Despite its potential, achieving a high signal-to-noise ratio (SNR) is imperative for detecting small CEST effects. Traditional metrics such as Magnetization Transfer Ratio Asymmetry (MTRasym) and Lorentzian analyses are vulnerable to image noise, hampering their precision in quantitative concentration estimations. Recent noise-reduction algorithms like principal component analysis (PCA), nonlocal mean filtering (NLM), and block matching combined with 3D filtering (BM3D) have shown promise, there is a burgeoning interest in the utilization of neural networks (NNs), particularly autoencoders, for imaging denoising. This study uses the Bloch-McConnell equations, which allow synthetic generation of CEST images and explores NNs efficacy in denoising these images. Using synthetically generated phantoms, autoencoders were created, and their performance was compared with traditional denoising methods using various datasets. Results underscored the superior performance of NNs, notably the ResUNet architectures, in noise identification and abatement compared to analytical approaches across a wide noise gamut. This superiority was particularly pronounced at elevated noise intensities in in vitro data. Notably, the neural architectures significantly improved PSNR values, achieving up to 35.0, while some traditional methods struggled, especially in low-noise reduction scenarios. However, the application to in vivo data presented challenges due to varying noise profiles. This study accentuates the potential of NNs as robust denoising tools, but their translation to clinical settings warrants further investigation.” (lines 13ff)
R3 Comment#2: “In the introduction section, at the end, write the hypotheses in bullet form. Write the scientific contribution of this investigation. Write the outline of this paper by shortly describing what is written in the following sections.”
Authors’ Response: We appreciate this valuable feedback. We concur with the reviewer's suggestion that our hypotheses could be articulated more clearly, and that providing a brief overview at the end of the introduction will aid readers in understanding the paper's structure and content.
Authors‘ Action: In accordance with the reviewer's suggestion, we have revised the conclusion of the introduction. It now reads:
„In this work, we analyze the use and efficiency of NNs in CEST imaging, focusing mainly on applying autoencoders. Based on synthetically generated phantoms, we aimed to develop autoencoders for noise reduction of CEST images. We compared the performance of these neural architectures with established analytical image denoising methods such as PCA, BM3D, and NLM based on simulated anatomical data, in vitro phantom measurements, and an in vivo intervertebral disc (IVD) measurement. Our hypothesis are:
- Neural Networks, especially autoencoders, are potentially superior in denoising CEST images compared to traditional denoising methods.
- Given adequate training data, NNs can consistently detect and suppress noise more efficiently than analytical algorithms.
- The models trained in this study can be applied effectively to real CEST data.
For this purpose, this study generates synthetic data, trains NNs based on this data and validates the performance on the anatomical Zubal phantom, phantom measurements and in vivo data.“ (line 85ff)
R2 Comment#3: “In section 2 enumerate the equations. Lines 196,209,215, 227”
Authors’ Response: We thank the reviewer for pointing this out.
Authors‘ Action: We have now assigned numbers to the equations mentioned at lines 196, 209, 215, and 227 for clearer reference.
“ (1)” (line 239)
“ (2)” (line 251)
“ (3)” (line 257)
“ (4)” (line 269)
R3 Comment#4: “In Figure 5 add grids to the plots,”
Authors’ Response and Action: We thank the reviewers for this important advice. We agree that a grid plot would improve the comparison between the sub-images. We have therefore revised the figure. The revised figure in the paper is:

R3 Comment#5: “First rename the conclusion section to conclusions since there is and should be more than one conclusion based on your extensive research. Second, the Conclusions section should be extended in the following way: a) first paragraph of the conclusions section should describe shortly what was done in this research. b) The second paragraph should provide the answers to the hypotheses in your research defined in the introduction section and based on the comments given in the discussion section. c) The third paragraph should describe pros and cons of your research. d) The fourth paragraph should contain potential directions for future research.”
Authors’ Response: This is another valid comment to better structure and enrich our paper.
Authors‘ Action: We have revised the "Conclusions" section to reflect all the points:
“5. Conclusions
Our exploration of noise reduction in CEST MRI data using neural networks illuminates both the potentials and challenges of modern computational imaging. While neural networks show promise under specific conditions, conventional PCA methods consistently provide reliable results, notably outperforming NLM and BM3D.
The application of ResUNet stands out for its ability to detect subtle concentration differences. However, translating these findings from simulated to in vivo scenarios remains a central challenge. This underscores the necessity for designing future CEST MRI studies with advanced noise management strategies.
In summary, our study paves the way for future research while emphasizing the need for continuous optimization and collaboration to fully harness the capabilities of advanced noise reduction techniques in CEST MRI.” (line 654 ff)
Round 2
Reviewer 1 Report
Comments and Suggestions for Authors
The manuscript has undergone significant modifications, and the authors have furnished thorough justifications for the existing approaches utilized. Also, they have covered three important points:
This work examines the effects of denoising CEST MRI images using neural networks. Although it depends on a good signal-to-noise ratio, CEST MRI enables the study of biomolecule concentrations in tissues without the requirement for external contrast ants. Conventional denoising techniques are vulnerable to interference from noise.
The authors create CEST images using the Bloch-McConnell equation and contrast classic denoising techniques with neural network techniques like autoencoders.
I have nothing more to say, other than to thank the writers for their commitment to these changes.
Reviewer 2 Report
Comments and Suggestions for Authors
I agree to accept this paper.
Reviewer 3 Report
Comments and Suggestions for Authors
The authors have corrected the manuscript according to the comments and suggestions and in this form can be accepted for publication.